# Entrepreneurial University Concept Review from the Perspective of Academicians: A Mixed Method Research Analysis

**Hüseyin Emre Coşkun [1,\*], Catalin Popescu [2,\*], Deniz Şahin Samaraz [3], Akif Tabak [3] and Bulent Akkaya [4]**

1 Department of Political Science and Public Administration, Izmir Katip Celebi University, Izmir 35620, Turkey
2 Department of Business Administration, Petroleum-Gas University of Ploiesti, 100680 Ploiesti, Romania
3 Department of Business Administration, Izmir Katip Celebi University, Izmir 35620, Turkey
4 Department of Office Management, Manisa Celal Bayar University, Manisa 45140, Turkey
\* Correspondence: huseyinemrecoskun@gmail.com (H.E.C.); cpopescu@upg-ploiesti.ro (C.P.)

**Abstract:** Understanding the role of the factors that contribute to the development and growth of entrepreneurial universities is important for both the university itself and the global economy as a whole. Therefore, this study aimed to gain a deep understanding of the entrepreneurial university. The study was designed with a mixed-method approach in which quantitative and qualitative methods were used together. In the quantitative part of the mixed-method research, a systematic literature review of the entrepreneurial university concept was done and based on the results of the literature search obtained by quantitative methods, general concepts and categories and their relations with the contexts were revealed. In the qualitative part of the mixed-method research, in order to explain the entrepreneurial university understanding, semi-structured interviews were conducted with academicians in universities. According to the findings, it was found that conceptualization was very important to understand the entrepreneurial university. The university was considered as an institution that serves students and companies via some channels, such as research reports, an academic publication, or a patent received at the end of a project. As the entrepreneurial university paradigm is being studied by various institutions in the world, this study contributes to the literature both as a theoretical study and as a study that reveal the concepts of the entrepreneurial university. By providing a conceptual framework, the research will contribute to the literature as a theoretical study by aiming to explain the service systems (universities, government, firms) based on the exchange of skills and the creation of common value in the knowledge-based society on the entrepreneurial university paradigm, which is extremely popular in the world. Findings from the systematic literature review and interviews with academicians were analysed comparatively, and a conceptual framework that evaluates the concepts and variables of the entrepreneurial university with a holistic view is presented.

**Keywords:** entrepreneurial university; competitive advantages; innovation; knowledge-based society; entrepreneurial university paradigm

## 1. Introduction

Over the past decade, the rapid change of competition depending on knowledge has shaped the relations of actors in the market [1,2]. Hence the need for expertise to create a competitive advantage in the marketplace turned to universities [3,4]. Firms have viewed the university as a source of human capital to fulfil their future employee demands up to now [5,6]. Over time, firms needed more qualified personnel from universities to gain the competitive advantage succeeding the effectiveness and efficiency. Nonetheless, the most crucial issue that leads firms to collaborate is that they have externalized their R&D activities to innovate their products and services outside of their factories and headquarters [7,8]. This breakthrough brought a new approach, so-called "open innovation" in the market, and

thus the need to externalize companies' R&D activities [9] has brought firms closer to the universities. In the eyes of the firms, universities are considered not only as of the source of human capital but as institutions where they co-create value [10,11] of their products and reduce research and development costs.

For this reason, the entrepreneurial university literature aims to ensure that the economic and social outcomes of the processes created by these three actors are the most efficient for all by examining the cooperation of public, industry and universities at micro, medium and macro scales. Triple helix Entrepreneurial universities, which play a distinctive role in the regional, national, and international arena, have a strong identity as a source of renewed trust between the society and the university [12]. The increasing role of knowledge in terms of society and the university in terms of the economy can be analysed with the phenomenon of the Triple Helix, which represents the university-industry-state relations. The relevant helix has been formed on the axis of university, industry, and government as intertwined rings with the motive of promoting academic research and economic development.

The product of this distinguished, the entrepreneurial university transcends the primary academic missions of education and research. Particularly, the mission of economic development is to contribute increasingly to the systematic production of scientific innovation and the re-establishment of the knowledge base [13]. As was stated already, public university-industry relations are defined in three ways [14]. According to them, these relations come after each other chronologically:

- the first model shows the relationship in which the nation-state includes academia and industry and manages the relations between these two sectors;
- the second model denotes a system in which the state plays a less dominant role, the boundaries of each of the institutional areas are clearly defined and their relations with others are outlined;
- the third model is a development and innovation model where three constantly evolving domains complexly interact with each other at different levels flexibly and develop evolutionary pathways that achieve not only their own goals but also the finalized goals of the other two entities. This model, known as the triple helix model, that regulates some public policy in Europe, has become a normative theoretical construct [15,16].

Since the triple helix model (Triple Helix) entered the academy, it has spread all over the world and the university has acted with the public and industry to contribute to local, regional, and national economies. This goal was expressed as the third mission [1,5,13]. In this sense, with a view to understanding the essence of university-industry-public cooperation, the first emergence of this idea should be sought from a historical perspective in the history of the university institution.

The first aim of this study is to determine the conceptual framework of entrepreneurial university studies through a systematic literature review. The subjects studied in the context of the entrepreneurial university will be determined by keyword analysis and co-occurrence analysis. In this way, a general picture of the literature will be formed. Then, network relations and density maps of these subjects will be prepared, and general concepts and categories of entrepreneurial university subjects will be expressed using the obtained findings.

In the qualitative part of the mixed-method research, to explain the entrepreneurial university understanding, semi-structured interviews will be conducted with academicians in universities.

By providing a conceptual framework, the research will contribute to the literature as a theoretical study by aiming to explain the service systems (universities, government, firms) based on the exchange of skills and the creation of common value in the knowledge-based society to the entrepreneurial university paradigm, which is very popular in the world.

## 2. Theoretical Background

Before all these fundamental changes, universities have already evolved into a new structure under some socio-economic conditions [1,5]. First, the history of the university as an institution should be recognized sufficiently then deepening these conditions that force universities into the new structure. The change experienced in universities is explained by classifying science-oriented medieval university (First Generation), education and research-oriented Humboldt-type university (Second Generation) and the new form that universities take today as an entrepreneurial and socially integrated (Third generation university).

The Middle Ages covers a period of about a thousand years, coming after the Greek-Roman culture and extending to the Age of Enlightenment. The Middle Ages is considered a transitional period between Antiquity and the Renaissance. Intellectual despair in medieval Western Europe led to the rise of learning centres, which were considered higher education institutions in many cathedrals and monastic schools after the Dark Ages. Although these schools were few in number, they could not fully develop until the 13th century and could not go beyond having a regional character. For a long time before the 13th century, there were no education and training institutions like universities in today's sense. Before medieval universities, there were churches, cathedrals, and monastic schools in Europe, as well as centres where private teachers taught a certain group. These centres were called *studium generale* [17].

The development of medieval Latin society enabled churches and cities to be organized independently, and the first universities also found their place in this executive organization. Universities that emerged in Europe in the Middle Ages differed from other educational institutions established at that time with their infrastructure, curriculums, rules, political-legal privileges, and extraordinary activities aimed at developing modern science. While the institutions where education and training were continued and knowledge production was realized until the 12th century in the Medieval Scholastic Period were the cathedral and monastery schools, in the 12th century, these schools lost their impact and their superiority to the universities. These new education and training institutions differ from the cathedral and monastic schools in the selection of teachers and students, regular curriculum, discussions, scientific studies, and libraries. On top of that, the books in which the knowledge is registered in the universities and mediate the dissemination of knowledge have become a tool for the production and dissemination of knowledge, not an icon that should be protected as it was in the early periods of Christianity. Universities were not established all at once in the Middle Ages. They emerged step by step. In the Middle Ages, universities, meaning completeness, was used for a series of cooperative associations. Therefore, this concept is used to state the association between teachers and students [18].

The University of Bologna, known as the oldest university in the world, was found in 1088. The University of Paris was founded in 1200, and Oxford, Cambridge, Arrezo, Palencia, Padua, Naples, and others began to be founded [19].

By the end of the 13th century there were about twenty universities and in the 14th century another twenty-five, among them the University of Prague. In the 16th century, many new universities were founded with the encouragement of reform and counter-reform movements. These first universities founded in the Middle Ages showed different characteristics in terms of both administrative structure and education systems, and with these features, they were a model for universities founded after them in Europe [20].

The university was founded in Bologna in 1088, and has ever since then passed through two substantial evolutions. The university became a research institution as a result of the initial evaluation [1,5].

Under the Humboldt principles of science, without abandonment of education role when Newtonian physics affected the whole of Europe, industrialization spread rapidly, nationalism and German idealism were on the rising idea of performing research in the university was born. So, the desire to reach absolute knowledge under the influence of philosophers such as Kant adorns the dreams of all European intellectuals [21,22]. Within

these facts, the German university system, the Humboldt Ecole, was born under the allowance of the King of Prussia and given his political protection and economic support.

The second evolution happened in the other continent, the USA, under the "land-granted universities" title. These universities were first built to solve the agricultural problems of the young established country. With the Morrill Act (1862), the land was given to the government and law-supported universities to increase agricultural production [14]. Following years with Bayh Dole Act, academics working at universities have been given to take licenses, patents, and copyrights from the outputs of their research and make profits from them [23–26]. After the practices of this law, the world's leading universities, the Massachusetts Institute of Technology (MIT) and Cornell University were also established.

In addition, with the asylum of German academics fleeing the Second World War, American universities also showed a different transformation and development in this research environment. Because of these changes, the number of licenses and patents in the region has increased substantially. This increase was also reflected in the region's economic prosperity and created a unique production, art, and business environment such as Silicon Valley.

Therefore, universities have transformed from the idea of being independent research institutes [1] to a structure that can respond to the needs of the socio-economic life around them. Service-dominant logic, which is the basic theoretical framework underlying this research, proposes a new perspective on the use and value of resources as opposed to the classical understanding of economics [10,27]. The concept of goods expressed in the classical understanding of economics describes tangible products [10]. Besides, service-dominant logic tries to make the activities of economic activity through both tangible and intangible products [10,27]. For this reason, as introducing service-dominant logic, Lusch and Vargo introduce both dominant goods and intangible services into the product conceptualization [28]. Thus, in the ICT age and value-based economic understanding, the needs of market actors based on information technologies are increasing. The need for information technologies and the professionals to use them in this new complicated world has made universities a part of this economic system in as much as companies are obliged to maintain their viability and sustainability through research and development in this direction [5]. Owing to this fundamental change university have transformed into service institutions. The increasing needs of the actors in the market on information technologies, companies' sustainability, and professionals for competition forcing companies to innovate forced universities to transform. Today, entrepreneurship is increasingly understood as a set of competencies needed for many professions to be integrated into education and training. This tendency towards an understanding of entrepreneurship in a broader way and, at the same time, a growing need for inclusion of sustainability issues in education, should consequently lead to the expansion of the entrepreneurial university concept. The postulated expansion of an entrepreneurial university approach creates a new response to some of the concept's criticism about the perceived growing dependency of higher education institutions from the industrial and business partners or sponsors [29]. Entrepreneurship education in entrepreneurial universities increased students' motivation and recommended tutor training with a real-world approach. It was also regarded as necessary to replace traditional exams with entrepreneurial assessments and link the knowledge acquired with other subjects in the curriculum and other entrepreneurial ecosystem actors [30].

In order to identify the skills necessary to create a qualification suitable for future entrepreneurs and to generate an appropriate framework from an educational point of view (including through the development of training programs), extensive studies were carried out at local, regional, national and international levels [31].

## 3. Research Questions

It is seen that many studies have been carried out with different levels and samples on both the definition of the entrepreneur (research) university concept and the revealing of its components in the country and abroad. For all that no study has been found in which

academicians with experiences such as consultancy in companies or entrepreneurship in techno parks have been researched with quantitative and qualitative methods. To meet this need, the basic research question was determined as if it is possible to understand the concept of an entrepreneurial (research) university from the perspective of academicians with such qualifications. The sub-research questions related to this research are as follows:

- Is it possible to reveal the entrepreneurial (research) university approach, which came to the forefront in the academic world in the 2000s, considering the theoretical changes it went through, in which contexts it is studied in the literature, the main themes and the concepts with which they are studied, with the help of network analysis, density analysis and coding co-occurrence tables?
- With semi-structured interview questions created with the help of these concepts, which were revealed by taking expert opinions, how can implicit knowledge, which is obtained from in-depth interviews with academicians who had experiences such as consultancy in companies or entrepreneurship in techno parks, be incorporated into entrepreneurial (research) university components?

### 4. Material and Methods

#### 4.1. Research Design

It is aimed to test the research questions of the project with a mixed approach in which quantitative and qualitative methods are used together. In this respect, the method section can be summarized in three stages in general. These stages are as follows:

- data processing phase and knowledge discovery phase, in which the main components of the entrepreneurial university literature are revealed through analysis and focus group studies, with the findings obtained from the network analyses of the articles in the Web of Science Core Collection between 1975–2017 based on the keywords taken as basis in the literature;
- the field research phase with interview questions based on the obtained research university components;
- the last stage is where the components in which the code-occurrence models obtained from the literature and the code-occurrence models obtained from the interviews are compared and where the study is concluded by determining components in which the entrepreneurial university concept is evaluated by evaluating the different practices/understandings on institutions and understandings.

#### 4.2. Data SOURCE and Analysis

In the first step of the research, basic concepts of the entrepreneurial university topic were obtained through a systematic literature review. Extensive studies in the field of entrepreneurial university were analysed. One such study [32] made a significant contribution to the entrepreneurial university sector since it analyzed a number of 420 articles published in the Web of Science database from 1972 to 2015 but added also an important contribution to the universal literature. From this point of view, the methods and principles used by Schmitz et al. (2017) were followed to determine the sample to be used in the systematic literature review.

Accordingly, Schmitz et al. (2017), in their work in the Web of Science Core Collection, articles with keywords *"universit* innova*"* or *"innova* universit*"* or *"universit* entrepreneu*"* or *"entrepreneu* universit*"* or *"academic* innova*"* or *"academic* entrepreneu*"* were detected. In the literature review section, following the methodology of Schmitz et al., the articles published in the Web of Science Core Collection between 1975–2017 were searched. These searched articles containing the following keywords: *"universit * innova *"* or *"innova* universit *"* or *"universit * entrepreneu *"* or *"entrepreneur * universit *"* or *"academic * innova*"* or *"academic * entrepreneur *"* [32]. Because it can be said that the way to reach sufficient data in the research depends on the correct selection of keywords.



In another comprehensive study, the stages suggested in mixed method applications were considered [33]. Hesse-Biber expresses that a research design can be done concerning mixed-method studies based on the following steps:

1—Preparation of research questions;
2—Application of research methods;
3—Data analysis and interpretation of data;
4—Formation of a new set of research questions;
5—Application of methods;
6—Data analysis and interpretation.

The research questions were restructured (step 3) and additional research questions were added (step 4) after the results from the bibliometric analyses were interpreted (step 2). The detailed steps of 1-2-3 stages that make up the quantitative part of the research are as follows:

- the articles about the entrepreneurial university were obtained through a systematic literature review. In the literature review section, following the methodology [32], the articles published in the Web of Science Core Collection between 1975–2017 were searched and themes and concepts of the entrepreneurial university approach were revealed in 617 articles;
- basic components of entrepreneurial university were determined based on the indexes produced in the country and abroad;
- the components of entrepreneurial (research) university paradigm were determined and obtained by density-network analysis with getting expert opinions in focus group studies, co-occurrence, and code-occurrence model analysis model.

Thus, at the end of the first phase of the research (data processing and information discovery phase), interview questions were prepared to be used in the field research, which is the second phase, and these questions were applied to academicians who had experiences such as consultancy in companies in the state university or entrepreneurship in techno parks. Nine academicians were contacted and six interviews were held at three different institutions due to various obstacles.

In this respect, the quantitative part of the research consists of 617 articles obtained from the literature, and the qualitative part consists of six interviewees.

In the last stage of the research (concluding by dividing into components), thematic analyses were made by coding the interviews with the concepts obtained from the systematic literature review in qualitative data analysis program MAXQDA. Subsequently, the concept of entrepreneurial university was evaluated by comparing the code co-occurrence models obtained from the literature with the code co-occurrence models obtained from the interviews.

Qualitative steps suggested in different sources were considered in the research [34]. According to this:

Step 1—First job with text: Memos (notes) were created by reading the deciphered interviews;

Step 2—Main topic categories were determined;

Step 3—First coding phase: The interviews were coded based on the main categories determined in step 2;

Step 4—Compilation of passages assigned to the main category: Thematic matrices were created using the sections coded with the main categories and summaries of the interviews were made;

Step 5—Identifying subcategories: Instead of sub-categories, a code set was created with concepts from the literature and new concepts to be added by the author;

Step 6—Second coding phase: All interviews were coded with the codes determined in step 5;

Step 7—Category-based analysis and presentation of results: The quotes that the interviewees said about the determined category were evaluated through thematic matrices and

the codes in which determined categories occur together were explained with quotations from the interviews.

As stated above, in this research, revealing the components related to the entrepreneurial (research) university topic is structured in a way that confirms each other with a mixed model by integrating qualitative and quantitative research methods. It is thought that bringing out all the discussions on the concept of entrepreneurial research university through scientific articles and including these concepts in interviews with expert opinions increases both the inclusiveness and originality of the study.

## 5. Results

### 5.1. Findings of the Literature Review

Before any remark, it is worth presenting, in summary, some examples of the most concrete and recent referential results and valuable contributions brought to the analysed field:

- the need to focus on entrepreneurial process competencies but also the usefulness of acquiring them, for the development of entrepreneurship learning progression [35];
- the need to train students in order to face the main difficulties that the labour market and the social component are proposing to the educational environment [36];
- the need to develop useful tools for assessing entrepreneurial skills that any interested person can accumulate during the training related to the educational period (pre-university and university) [37];
- the construction of an entrepreneurial university requires an important governmental financing but also measures to support innovative activities, without neglecting the teaching and research components [38].

Quantitative findings from the literature are important for this research. With the data obtained, a comprehensive approach to the entrepreneurial university literature has been developed. In order to make sense entrepreneurial university paradigm through the literature, 617 articles published between 1975 and 2017 on the Web of Science website were determined and the data obtained were subjected to various analyses to answer the research questions. 617 articles were downloaded in .txt format, including all the information, and examined in VOSviewer program. Data from Figures 1–6 were obtained from this analysis using the VOSviewer network analysis program. In addition, the results obtained from the systematic literature review were also used in the coding of the in-depth interviews.

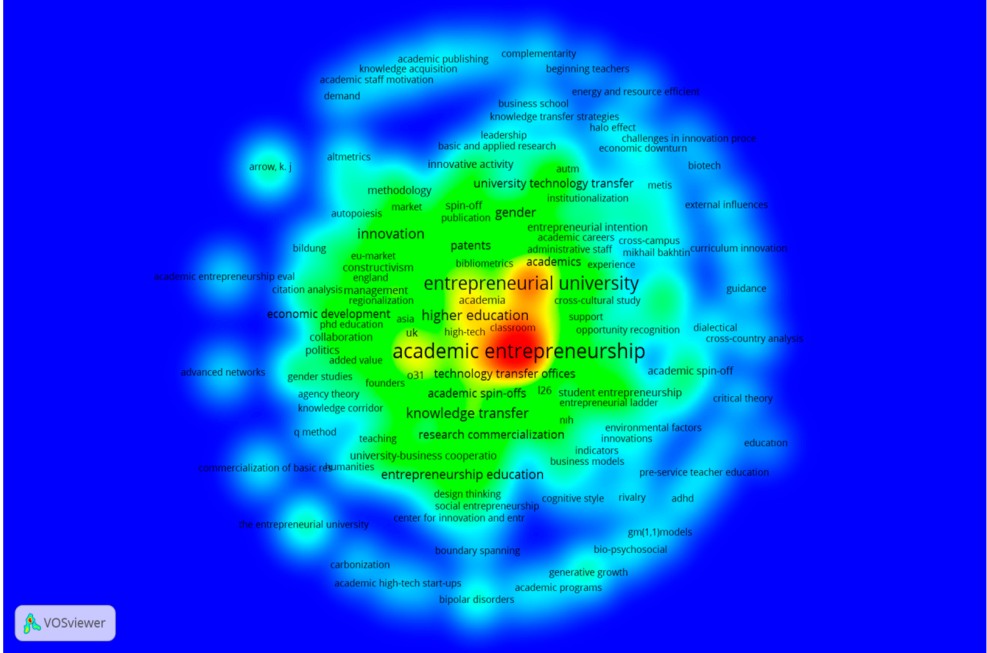

**Figure 1.** Density Map of 1145 Keywords with Co-occurrence Method.

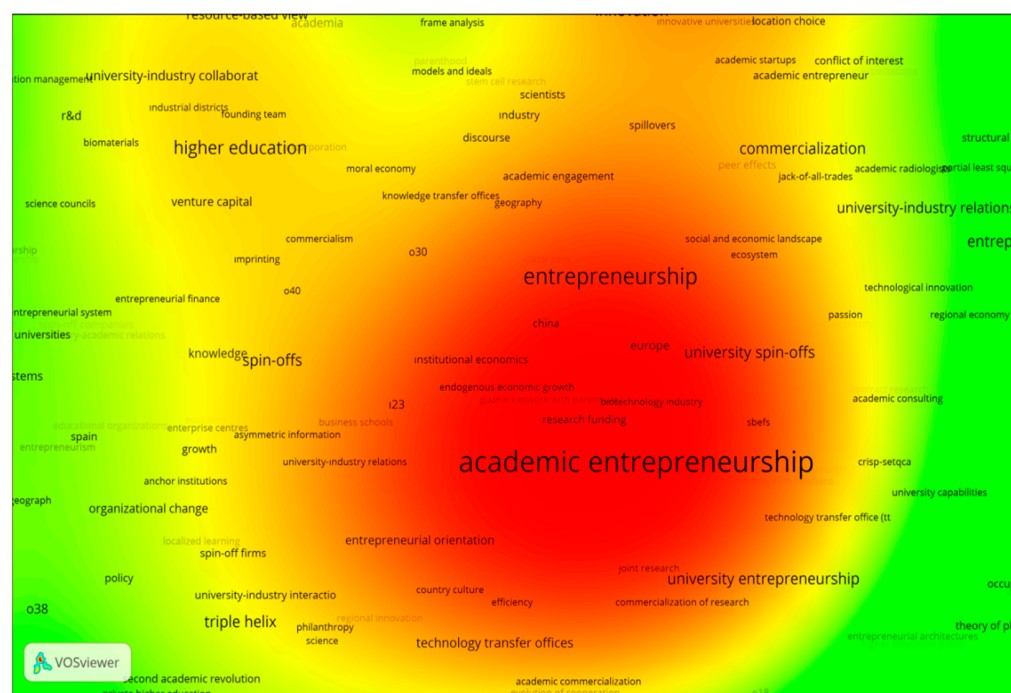

**Figure 2.** Density Map of 1145 Keywords with Co-occurrence Method (a zoomed-in representation).

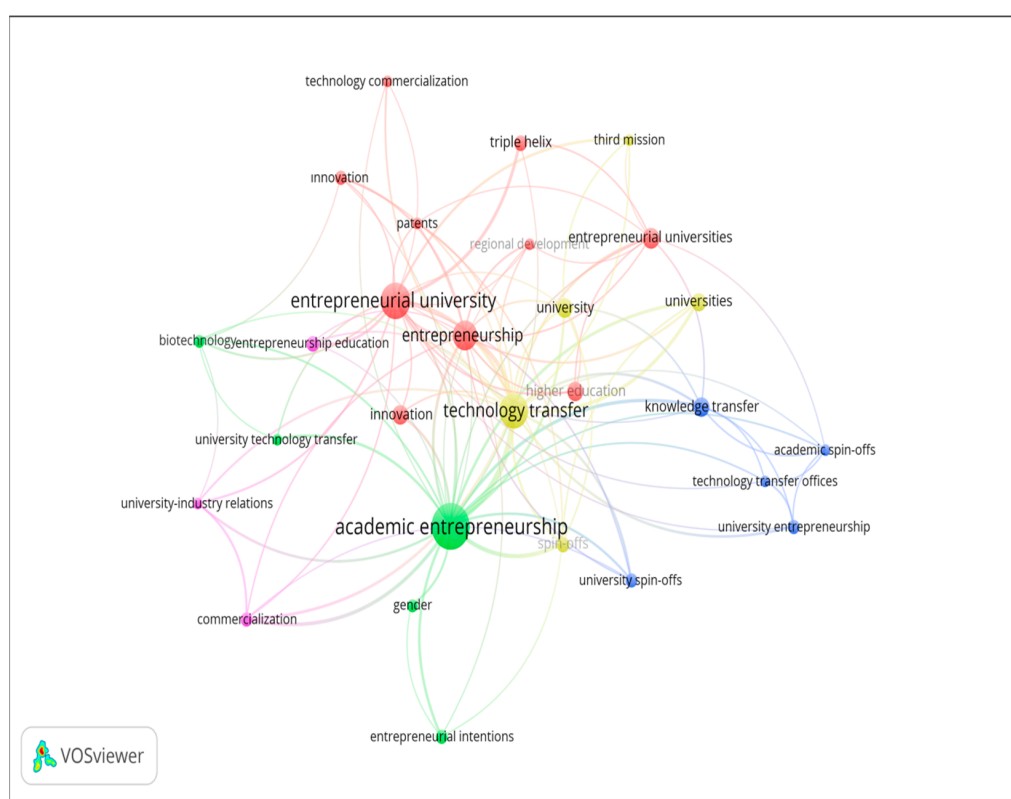

**Figure 3.** Network Map of most used 28 Keywords with Co-occurrence Method.

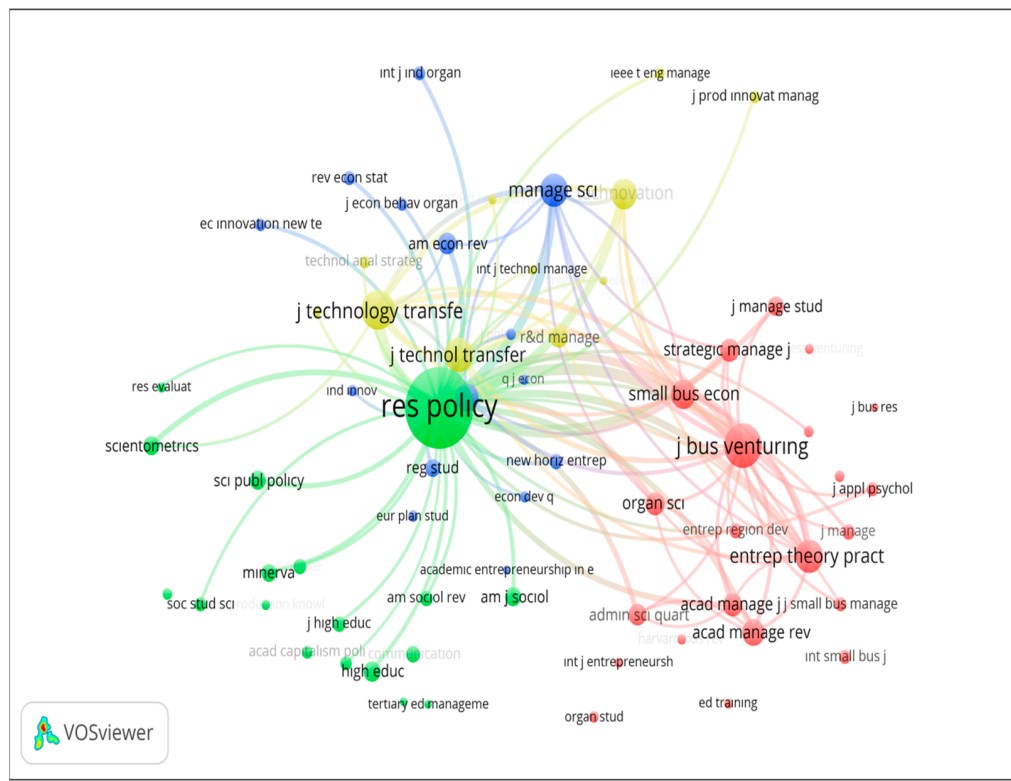

**Figure 4.** Network analysis of 65 journals with Co-Citation Method.

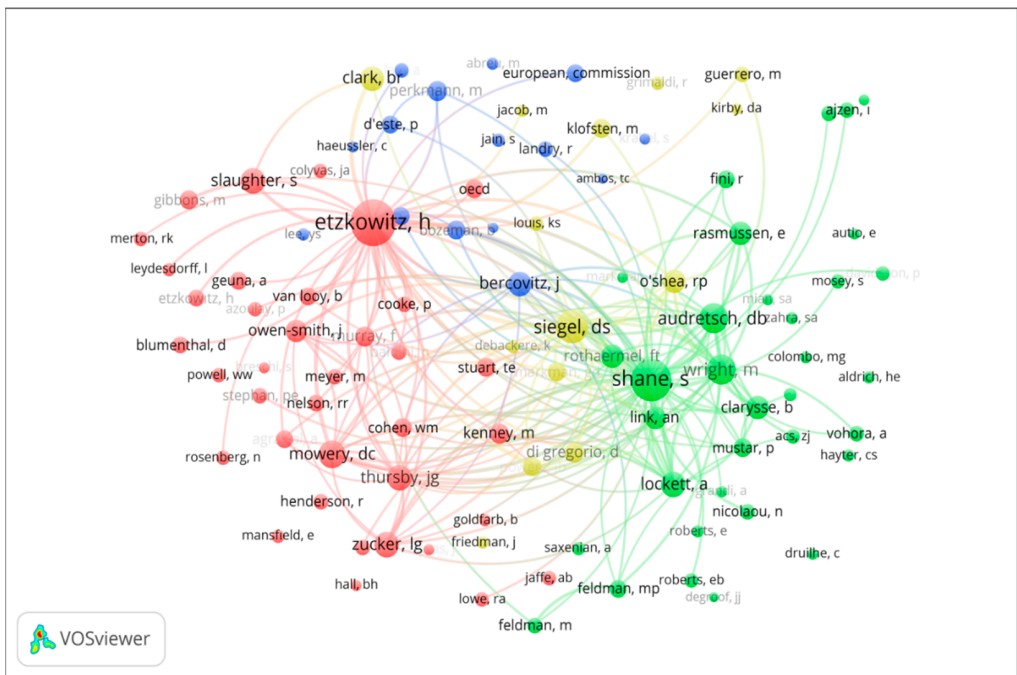

**Figure 5.** Network analysis of the most productive Top 100 authors with Co-Citation Method.

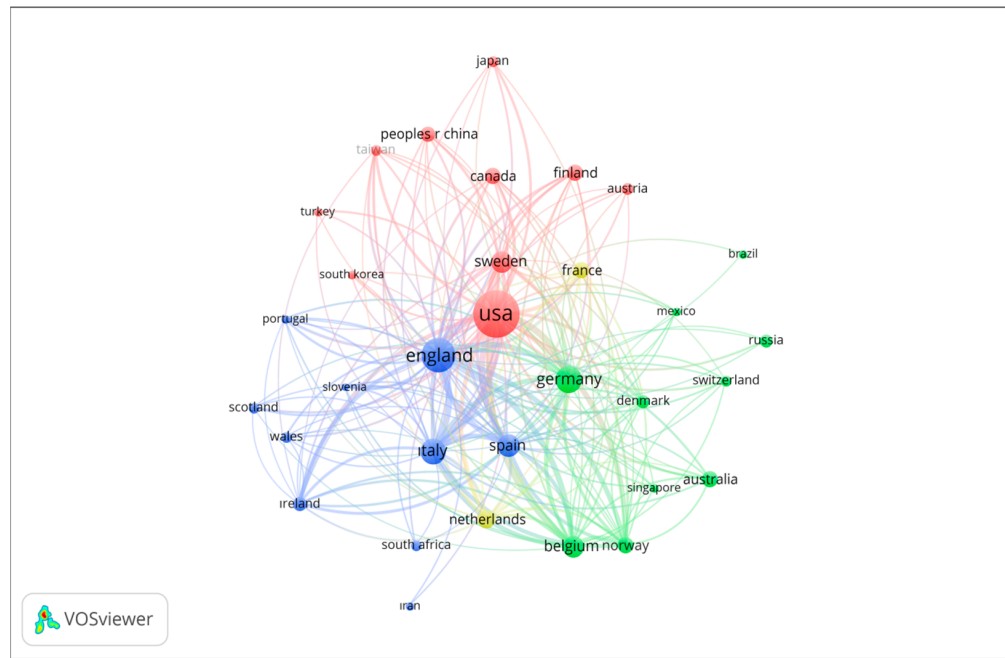

**Figure 6.** Network analysis of the 32 countries with Highest Bibliometric Coupling.

Figure 1 shows the centre of the density analysis. According to Figure 1, the literature is centred around the "academic entrepreneurship" term and other concepts such as research funding, technology transfer office, open science, commercialization, university students, organizational change, technology transfer, regional innovation, collaborative research, joint research, university-industry associations. Even though the concept of open innovation is from the development of the milestones that prepare the formation of the entrepreneurial university paradigm, it has been found not to have an important role in the literature. Even so, it can be said that alone the 'innovation' concept is among the most used keywords.

Henry Etzkowitz (2003) emphasizes the academy' internal and external connections and ability to transfer information and technology to commercial firms [39]. The concepts he mentioned also appear in Figures 1 and 2.

In order to get a better picture for the paper findings in relation with the keywords searched, is presented below in Figure 2 a similar density map with the Figure 1, but with a zoom-in representation.

In order to understand how the map can be read, in the following paragraphs are made few statements. For instance, to understand the concepts such as "research funding", the following questions were asked to the participants during the interviews as stated in the third step of the qualitative part of the research design: "Did you get income/ salary/scholarship for your research?", "Who paid this money?", "Was it satisfactory from your income/money?", "Was your income/money fair for you?".

On the other hand, to understand the concept of "open science" was generated the question: "During the research, did you have access to all kinds of materials (books, articles, labs, etc.)?".

To understand the concept of "joint research" and "collaborative research" was designed the following question: "In what kind of cooperation did you or your co-workers have in this project?".

To make sense of the concepts of "academic entrepreneurship" and "entrepreneurship activity" the question: "In your opinion, what is the role of academicians in this new university concept?" was asked.

A new step was to generate a network map by taking into account the most used keywords from the searches (Figure 3).

As seen in Figure 3, the entrepreneurial university literature is clustered into five main groups. These groups are described by some key points, as follows: entrepreneurial university (representing the red cluster), technology transfer (representing yellow cluster), knowledge transfer (representing blue cluster), academic entrepreneurship (representing green cluster) and commercialization (representing purple cluster). These clusters key points were considered as the five main categories of studies in literature as an outcome of the systematic literature view of this research, and they will be coded as the themes at the initial coding stage of the in-depth interviews.

Rothaermel et.al. (2007) study also provides confirmatory explanations for these five clusters. This article emphasized that universities must interact with the industry through technology transfer and this interaction is one of the main factors triggering the concept of the entrepreneurial university [4]. Kogut and Zander (1992) emphasize that technology transfer plays an important role and is necessary for the growth of companies [40].

Another representation is based on the Co-Citation Method (articles about entrepreneurial university) and describe a network analysis in relation to a significant number of relevant journals in the field (Figure 4).

As can be seen in Figure 4, there are generated three main clusters (with green, red and yellow colour). It can be said that a significant part of the articles about the entrepreneurial university literature is quoted from publications in the *Research Policy* journal. Consequently, evaluating the network table, *Research Policy*, *Journal Technology Transfer*, and *Journal of Business Venturing* are the most central journals in the literature.

In Figure 5, which contains also a network analysis, the first 100 authors were envisioned using the co-citation analysis method. According to Figure 5 it can be stated that Etzkowitz H, Shane S and Siegel D.S form three main centres (names) in the specific literature. In this way, the authors who founded the specialized literature in the field and prepared and introduced the foundations and reference topics were established. As well as that, examining the authors' co-occurrence models are also significant in terms of expressing the academic collaborations of these writers with their colleagues.

In respect of this analysis based on the bibliometric coupling method, countries that carry out similar studies on the subject and collectively work with each other were examined. Publication clusters which share similar references and schools were obtained. These clusters can be divided into three major groups of countries: first cluster represented by continental European countries such as Britain, Italy, and Spain, second cluster containing countries such as the USA, Sweden, Canada, China, Japan and the third cluster represented by Germany, Belgium, Denmark, Switzerland, Australia, Singapore, and Mexico.

*5.2. Findings of the In-Depth Interviews*

In-depth interviews were coded and analysed in the MAXQDA program. Much as in the literature review, maps of the co-occurrence of concepts have been drawn. At the same time the differences between our sample and the world literature have been compared.

Entrepreneurial university paradigm practices are starting to be implemented in state universities in Europe. In consideration of information obtained from the interviewers, it is understood that this transformation is not an institutional quite the contrary an individual transformation. It can be said that the individual interests and curiosity of academicians have a determinant role in the realization of local and national programs aiming at entrepreneurial activities.

It is understood that regional cultures and social life are associated with the university. According to the interviews the research funds provided by state university are not well utilized.

Interviewers references less to the concept of commercialization in the entrepreneurial university or the innovative university approach. In particular, the interviewees did not give any answer regarding the profit expectation. The results of the study by Göktepe-Hulten and Mahagaonkar [41], which revealed that the aim of German academicians is not to be profit-oriented, were confirmed in our sample. Interviewers explained that

commercialization is generally perceived as 'an economic benefit of knowledge'. That is to say, the part that makes a profit is not the job of the academician, but the job of the companies. The point that academicians emphasize is the concept of "conclusion", which takes the first place in the models of coexistence. As a result, researchers have frequently used expressions that may correspond to an understanding that they can transform their research into practical life.

Academicians had no difficulty accessing any source, journal, or laboratory in their research and teaching activities. Because these three institutions have a wide range of articles and database networks that follow the principles of "open access".

'Problems encountered during research' code is bureaucracy, the content of financial funds (in some cases, not meeting the items such as transportation) and cultural reasons and these have high importance in our sample.

The interviewees were asked, "What can be done to improve their research, taking into account their experience?" and often different answers were received. These answers were such as working with more research centres, facing fewer bureaucratic problems, more teamwork and more intercultural work.

The question "*Do you feel yourself in the decision-making mechanism of your university?*" received two yes answers. In one of these answers, the interviewee stated that he tried as hard as he could, while the other stated that he considered joining various research groups as a part of the decision mechanism. Academics who do not feel themselves in the decision-making mechanism stated that they are not in such a position due to their position and functions.

The interviewees are divided on the financing of the university. This dilemma emerged as the university is financed by the private sector or the state. The main argument used by the interviewees advocating continuing government funding is research and theoretical work will be overshadowed. In addition, academicians must have the identity of 'researcher', unlike private sector employees. Also, academics advocating private sector university financing stated that companies are more result-oriented and move faster. Results-oriented approaches have shown that research has a positive impact. Another critical issue that needs to be tackled in this consideration is that state universities are far from entrepreneurial approaches and academics reject the concept of management designed from the ground up.

Academicians want to work with private sector employees instead of public sector employees. The reasons for these choices are to conclude their research, see them in practice, work faster, seek the truth rather than theoretical-oriented issues, and share the international environment provided by the private sector. An interviewee also stated that there are some problems in communication with a higher level in the public sector compared to the private sector.

One of the most important results of the research is that the majority of academicians view interdisciplinary, multidisciplinary and intercultural research positively. They also mentioned that without these qualifications, even research is worthless and useless. It has been stated that it is difficult to produce something new for an academic who constantly researches the same subjects. It has been stated that research processes can be instructive for academics, even if no results are obtained in other disciplinary studies.

In literature, the entrepreneurial university concept is studied separately in different contexts with a wide variety of methods, and the lack of a conceptual framework to evaluate concepts and variables of the entrepreneurial university with a holistic view is emphasized.

## 6. Conclusions

Universities have undergone great changes until they reached today's university understanding. While raising professionals within the framework of the first university approach, science was their focus. Afterwards universities have become institutions where the research principle is the main subject. This structure, also called as Humboldt École, enabled universities to gain an institutional structure.

The university model in which universities do only research has evolved with industry collaborations. In this model, which is called the third generation, in addition to classical features such as education and research, universities have begun to play an active role in the socio-economic development of society.

In the first step of the research, basic concepts of the entrepreneurial university were obtained through systematic literature review. Systematic literature review was aimed to reveal the conceptual framework of entrepreneurial university literature. In the first step, based on the results of the literature search obtained by quantitative methods, general concepts and categories and their relations with the contexts were revealed. In the qualitative stage of the research, semi-structured interview questions were prepared by taking these quantitative data into consideration and the answers to the research questions were sought through the selected sample. the results obtained in the systematic literature review were used for the coding of in-depth interviews. Interviews with in-depth interviews were coded and analysed in the MAXQDA program. In the light of information obtained from the interviewers, although there are local and national programs aiming at entrepreneurial activities, it can be said that the individual interests and curiosity of academicians have a decisive role in the realization of these programs. Interviewers have touched little on the concept of commercialization in the entrepreneurial university, or in other words, the innovative university approach. When we look deeply at these findings, which can be considered as one of the remarkable results of our study, it can be said that commercialization is generally perceived as 'an economic benefit of knowledge'. In other words, the profit-making part is not the job of the academician but the business of the firms. The point that academicians place emphatically on is the concept of "finalization" that comes first in the models of co-occurrence. Accordingly, researchers often used expressions that may correspond to an understanding that they can turn their research into the practical life.

To this research, it was seen that the classical functions of universities were not sufficient. Universities have become more sensitive to market demands and have started to serve in a model that generates income other than public resources. The entrepreneurial university incorporates the teaching and research academic models and takes them to the next stage of development, integrating forward and reverse linear models into a renewed 'social contract' between the university and the larger society, for creating economic and social enterprises as the quid pro quo for large-scale funding of the academic enterprise [42].

As in every process, there are some difficulties, such as management weaknesses of the state-industry-university, insufficiencies in funding, inadequacy of state-industry-university mechanisms, encountered here as well.

For the development of the entrepreneurial university and its settling in society, the triple helix model approach needs to be actively adopted by university, industry, and state. As so, entrepreneurial universities will be able to respond to expectations about providing added value to the economy in their regions.

**Author Contributions:** Conceptualization, H.E.C., C.P., D.Ş.S., A.T. and B.A.; Data curation, H.E.C., C.P., D.Ş.S., A.T. and B.A.; Formal analysis, H.E.C., C.P., D.Ş.S., A.T. and B.A.; Investigation, H.E.C., C.P., D.Ş.S., A.T. and B.A.; Methodology, H.E.C., C.P., D.Ş.S., A.T. and B.A.; Supervision, H.E.C., C.P., D.Ş.S., A.T. and B.A.; Validation, H.E.C., C.P., D.Ş.S., A.T. and B.A.; Visualization, C.P.; Writing—original draft, H.E.C., C.P., D.Ş.S., A.T. and B.A.; Writing—review & editing, H.E.C., C.P., D.Ş.S., A.T. and B.A. All authors have read and agreed to the published version of the manuscript.

**Funding:** This research received no external funding.

**Institutional Review Board Statement:** Not applicable.

**Informed Consent Statement:** Not applicable.

**Data Availability Statement:** Not applicable.

**Conflicts of Interest:** The authors declare no conflict of interest.

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
