# Peer review of "Entrepreneurial University Concept Review from the Perspective of Academicians: A Mixed Method Research Analysis"

_sustainability, doi:10.3390/su141610110_

Round 1
Reviewer 1 Report
Abstract
You could set the aim of the paper and the results more explicit.
Introduction
Maybe is better to present the structure of the paper at the end of this section, not at the end of Theoretical Background.
Theoretical background
More papers from recent years and also from Sustainability journal must be analyzed in this section.
Material and methods
In the Data source and analysis section be more quantitative, number of articles, number of academicians, period of interview…
Result
You must explain in a quantitative manner how data from figure 1-6 were obtained.
Also, you must specify how many articles published in the Web of Science Core Collection were analyzed, and how many academicians were interviewed and how were they selected
Discussion
In the discussion section the results must be compared with other similar study results.
Conclusions
Ok
Author Response
Dear Reviewer,
The authors wish to thank to the reviewer for the time and effort in reviewing the manuscript, as well as for the suggestions and comments. We appreciate a lot your comments! These things improve the content and the value of the research itself. We attached a file containing all responses, changes and added parts, as representing our reply to your comments, remarks and suggestions. Also there is an updated paper version.
Thank you!

Reviewer 2 Report
Thank you, I find the article interesting however I have the following recommendations:
1) I suggest supplementing the bibliography to give a perspective on the concept of the entrepreneurial ecosystem. While showing how the topic has evolved, it is suggested to include more recent literature that discusses this approach. Then, I suggest including and reviewing literature from these articles:
Portuguez Castro, M.; Gómez Zermeño, M.G. Identifying Entrepreneurial Interest and Skills among University Students. Sustainability 2021, 13, 6995. https://doi.org/10.3390/su13136995
Portuguez Castro, M. Ross Scheede, C. & Gómez Zermeño, M. (2020).Entrepreneur profile and entrepreneurship skills: Expert's analysis in the Mexican entrepreneurial ecosystem, 2020 International Conference on Technology and Entrepreneurship - Virtual (ICTE-V), San Jose, CA, USA. https://ieeexplore.ieee.org/document/9114372
2) I suggest reviewing the approach, rather than a mixed study it is a literature review that could be a systematic literature review or justify the use of mixed methods for this purpose.
3) Add a figure in which the criteria for exclusion and inclusion of articles can be observed.
4) Can you add a link to the data base?
5) Reorganize the results better according to the research questions, as well as put the main citations from which the data arise.
Although I can see that work has been done on the literature analysis, more work still needs to be done on the categories obtained from the research questions. An SLR should include at least some of the authors found in the literature and show quantities of articles; the authors should rework the information obtained to be of more interest to the reader.
I suggest looking for examples of systematic literature reviews to identify how information is presented in this type of study.
Author Response

(The authors gave the same response as above.)

Round 2
Reviewer 2 Report
Thanks for the clarifications, it is still necessary to include the most important citations in the results. For example for each figure include which are the studies that talk about the main topics.
Author Response
Dear Reviewer,
The authors wish to thank to the reviewer for the time and effort in reviewing the new version of the manuscript, as well as for the suggestions and comments. We appreciate a lot your comments! We attached a file containing our response, changes and added parts, as representing our reply to your new comments and suggestions. Also there is an updated paper version (new revised version).
Thank you again!
